The impact of sanctuary visits on children’s knowledge and attitudes toward primate welfare and conservation

Feliu Olga o.feliu@fundaciomona.org 1 2
González-Zamora Arturo 3
Riba David 2
Sauquet Teresa 2
Sánchez-López Sònia 4
Maté Carmen 5
1 Department of Clinical Psychology and Psychobiology, Faculty of Psychology, University of Barcelona , Barcelona , Spain
2 Research Department, Fundació Mona , Girona , Spain
3 Instituto de Investigaciones Biológicas, Universidad Veracruzana , Xalapa , Veracruz , Mexico
4 Department of Psicobiology, Universitat Oberta de Catalunya , Barcelona , Spain
5 Department of Animal Rights, Barcelona City Council , Barcelona , Spain
Vonk Jennifer
Electronic publication date: 2023 Jun 16
Publication date: 2023
Volume: 11
Electronic Location ID: e15074
Received 2022 Oct 4; Accepted 2023 Feb 23
Copyright: ©2023 Feliu et al.
Copyright year: 2023
Copyright holder: Feliu et al.
License: This is an open access article distributed under the terms of the Creative Commons Attribution License, which permits unrestricted use, distribution, reproduction and adaptation in any medium and for any purpose provided that it is properly attributed. For attribution, the original author(s), title, publication source (PeerJ) and either DOI or URL of the article must be cited.
License URL: https://creativecommons.org/licenses/by/4.0/

Keywords: Conservation, Education, School children, Assessment, Primates, Sanctuary, Attitudes, Knowledge, Wildlife trade, Chimpanzees

Funding: The authors received no funding for this work.

==============================
Primate sanctuaries provide a solution for the increasing number of primates being taken from their home countries to support the demands of the illegal pet trade. To help end the primate trade and raise awareness about the risks this trade poses to delicate ecosystems, sanctuaries are increasingly developing conservation education programs. Education and raising awareness must be one of the primary roles of primate sanctuaries. However, there are few evaluations of the impacts of conservation education programs for school children published in scientific literature. To address this gap, we conducted an evaluation of educator-led visits of school children at Fundació Mona, a primate sanctuary located in Catalunya, Spain. Questionnaires for an experimental and control group were conducted with 3,205 school children, ages 8 to 18 from 83 different schools, to evaluate changes in their attitudes and knowledge of primate welfare and conservation. We found that Fundació Mona’s program of environmental activities had a positive impact on children, both female and male students, in terms of attitudes and knowledge of primate welfare and conservation. Although female students gave better responses regarding welfare and conservation, all children showed gains in pro-conservation responses. This study demonstrates that environmental education activities focused on children can help shape a change in knowledge and attitudes toward primate welfare and conservation. Educator-led visits of school children to primate sanctuaries such as Fundació Mona can also serve to amplify biodiversity conservation messages among children and their families. We encourage primate sanctuaries to promote empirical studies of attitudes and knowledge of primate welfare and conservation and to conduct systematical evaluations to strengthen their educational activities.

Introduction

The primary goal of primate sanctuaries is to provide a safe and nurturing environment for primates who have been rescued from the illegal pet trade (Farmer, 2002; Ferrie et al., 2014), research laboratories (Lopresti-Goodman, Bezner & Ritter, 2015; Fultz, 2017), or other harmful situations (Feliu et al., 2022). Many sanctuaries are developing and carrying out conservation education programs to support efforts not only to end the illegal primate trade but also to raise awareness regarding the risks this trade poses to primates’ endemic ecosystems (Kuhar et al., 2012; Hansen et al., 2016) and promoting attitudes in society that favor primate conservation (Brent, 2001; Farmer, Buchanan-Smith & Jamart, 2006; Beck et al., 2007; André et al., 2008; Kuhar et al., 2012).

The main goal of most environmental education programs is to change participants’ attitudes to more sustainable and environmentally friendly behavior (Liefländer & Bogner, 2014; Jacobson, McDuff & Monroe, 2015; Esson & Moss, 2016). To achieve this, sanctuaries can adopt non-formal educational programs with school children fostering values of respect for animals and nature. For this reason, primate sanctuaries provide unique opportunities for children to learn about and connect with the primate world, and to develop an understanding of the importance of protecting them and preserving their habitats. Therefore, primate sanctuaries are key educational centers for environmental education programs and provide a platform for raising awareness of future generations and reinforcing environmental childhood education in the short, medium, and long term (André et al., 2008; Kuhar et al., 2012; Bowie et al., 2020).

Primates living at a sanctuary have names and usually tragic stories. Children engage emotionally with them, as they learn about their life histories. Their stories, though unfortunate, can promote a sense of connection with the rescued primates (Skibins & Powell, 2013) and as flagship species, the knowledge about their plight can contribute to the conservation of other taxa (Wich & Marshall, 2016) and increase positive attitudes toward conservation of the species and their habitat (Lukas & Ross, 2005). These types of positive experiences with nature and animals during childhood promote children’s commitment to protecting the environment when they become adults (Wells & Lekies, 2006; Chawla & Cushing, 2007). The knowledge and respect for an endangered species that children acquire when visiting a sanctuary can be also transmitted to adults (Rakotomamonjy et al., 2015). It is critical that young generations acquire respect for wildlife conservation as the future of many species will be in their hands (Schuttler et al., 2019).

Despite their important role in primate conservation and raising awareness, many primate sanctuaries tend to not have the necessary human or financial resources to run and assess conservation education programs for school children mainly due to the lack of funds. The educational and outreach value of sanctuaries has been poorly studied (Falk et al., 2007; Kuhar et al., 2010; Lukas et al., 2017). André et al. (2008) and Bowie et al. (2020) found four hundred Congolese children successfully acquired key knowledge about conservation after a visit to a primate sanctuary and that the information was retained after a second visit a year later. Kuhar et al. (2010) published data from the educational programs of other five sanctuaries in Africa showing stronger pro-conservation attitudes toward primates after visiting primate sanctuaries.

The zoo community, however, has a history of experience related to conducting and evaluating educational programs. Many studies have been carried out to assess the increase in knowledge and change in attitudes after visiting zoos (Ogden et al., 2004). Children after visiting zoos increase in learning outcomes (Randler, Kummer & Wilhelm, 2012; Jensen, 2014), in conservation-related knowledge and attitudes towards conservation (Moss, Jensen & Gusset, 2017b). Educational assessment programs have also been carried out at natural parks and biosphere reserves showing an increase in knowledge and positive attitudes toward the environment after completing the program (Kuhar et al., 2007; Rakotomamonjy et al., 2015). These findings are also supported by other studies done at sanctuaries (André et al., 2008; Kuhar et al., 2010; Kuhar et al., 2012; Grúňová et al., 2017; Bowie et al., 2020).

In order for environmental education programs in sanctuaries to evaluate their impacts, both within the framework of well-being and conservation, and improve their applicability in future initiatives, it is necessary to generate evidence from studies that provide solid and systematic data and have a significant presence at local and regional level in school children in the territory. The main goal of our study was to assess whether school children’s participation in an educator-led visit to the Fundació Mona Primate Sanctuary increased biodiversity conservation knowledge and fostered positive attitudes toward primate welfare and conservation. We also evaluated whether the grade level and gender of the participating children impacted the outcomes. To achieve our goals, we did two different studies. In Study 1, we used a questionnaire to assess children’s knowledge regarding primate species, their welfare and conservation after going through Environmental educational activities (EEA) at the sanctuary. We predicted that children going through EEA across grade levels of both genders will increase accuracy compared to the ones that did not participate in the EEA. We also predicted that gender and grade level of the participants will influence the responses obtained after going through the EEA. In Study 2 we used a questionnaire to assess children’s attitudes toward primate welfare and conservation. We predicted that children across grade levels of both genders will increase their pro-conservation responses after going through the EEA. We have included the variables of sex and grade since previous studies have found differences in the responses for these variables (Borchers et al., 2014).

Due to the shortage of educational studies at primate sanctuaries, the results of this study are important in encouraging other primate sanctuaries to develop and evaluate educational programs for school children. The results can also support the primate community, as program evaluation is important to developing best practices in environmental and conservation education (Jacobson, 1987; Bennett, 1989; Thomas, 2016).

Materials and Methods

We develop two studies with 3,205 school children between 8 and 18 years old to assess the knowledge, welfare and conservation of primate species and the attitudes toward primate welfare and conservation. The research, conducted with school children, was approved by the Board of Trustees of the Fundació Mona and the Bioethics Committee of the University of Barcelona.

Study site

The Fundació Mona (Mona) sanctuary is a primate rescue center located in the province of Catalunya, in the Northeast of Spain. It was founded in 2001 to provide a shelter for pet and entertainment primates smuggled into Spain from Africa. The primary goal of the Mona sanctuary during this time was to provide lifelong care for rescued chimpanzees (Pan troglodytes) and other primates, as well as to ensure a species-appropiate environment where they can live out the rest of their lives. The Mona sanctuary houses two groups of chimpanzees (Pan troglodytes) in an open environment of 5,640 square meters. It also houses a group of Barbary macaques (Macaca sylvana).

Fundació Mona began its education program for schools three years after its first chimpanzee rescue. In 2004 the sanctuary created an education department to develop educational activities for children ages 3 through 18. Since 2004, over 18,778 school children have participated in educational activities at the Mona sanctuary. In 2014 the education department developed the MonaEduca program, a new educational project dedicated to raising awareness of the dangers faced by primates around the world.

The MonaEduca program

In addition to raising awareness, the MonaEduca program aims to provide education related to primate welfare and conservation and to transmit the core values of respect for wildlife and for nature. The program aims to empower children to become agents of change for sustainability and to foster a society that is more respectful towards the environment. MonaEduca educational materials are designed to increase knowledge about primate species. Specifically, they educate about conservation issues in the wild, the illegal pet trade and the entertainment industry, and the role of sanctuaries in providing conservation support, and in rescuing, rehabilitating, and housing primates that have become victims of human activities. The program is carried out through a single school visit to the sanctuary that lasts approximately four hours. The curriculum targets all ages within primary, secondary, and high schools (8 to 18 years old). Educators use age-specific content for each audience that is delivered to groups with a maximum of 25 participants. The activities are always carried out by a guide/educator who has significant training in primatology and conservation. During the MonaEduca Program all children participate in three different environmental education activities (EEA) (Supplementary Material 1).

1. An indoor, introductory, dynamic, and participatory talk, where school children learn the objectives and roles of the sanctuary, how to behave in the presence of the rescued primates and information related to taxonomy, distribution, natural history, behavior, welfare and conservation.

2. An outdoor, educator-led visit where children observe the primates and learn about the story of each of the rescued chimpanzees and macaques.

3. A learning-through-play activity adapted to different age groups. The concept is that the children will participate in playing the role of every player involved in a conservation scenario. For example, they can play the role of the director of a palm oil company, a conservationist or a man from the local community working for a palm oil company because this is his only source of income which is essential to feed his family. This provides the school children participating in the MonaEduca activities with the knowledge and experience to protect the interests of every person involved in the conservation scenario.

Study 1: knowledge assessment

Knowledge assessment methods

In this study we developed and administered a questionnaire to obtain information about children’s knowledge about primate species, their welfare and conservation to assess whether school children participating in the EEA learned about primate species and primate welfare and conservation through a questionnaire with eight closed-format questions related to natural history, behavior, primate welfare and conservation (Fig. 1). The content of each question was presented during the educator-led visit. In each of the questions, children had to choose one option from among two, three, or four possible answers. Questions 2, 5, 6 and 7 addressed general knowledge about chimpanzee species, and questions 1, 3, 4 and 8 captured the participants’ knowledge about chimpanzee welfare and conservation. A total of 1,549 school children participate in this study. In terms of educational levels, 40% were in primary school, 57% were in secondary school and 3% were in high school.

Figure 1 Knowledge questionnaire.

The list of questions in the knowledge questionnaire.

After arriving at the Mona’s educational area, children were randomly divided into two groups: the control (CO) group and the experimental (EX) group. The control group completed the questionnaire without participating in the EEA, and the experimental group completed the questionnaire after participating in the EEA at the sanctuary. This study had a between-subjects design, so each participant made up one of the two conditions (control or experimental). This ensured that participants’ responses were not affected by being exposed to the same question twice (Clayton, 2017; Bowie et al., 2020). The questionnaire included a preliminary section on demographic details including gender, age, grade, and school name (Table 1). All responses were treated anonymously. The teachers who accompanied the school children who participated in the survey were informed of the goals of the survey before the children answered the questions. Children answered the questionnaire independently via a google survey using tablets on site. Six children at a time were able to answer the questionnaire, as we had six tablets placed at the entrance of the educational room. To measure the internal consistency of the questionnaire we used a Kuder-Richardson Formula 20 (KR-20) (Kuder-Richardson 20 Formula, 2014). The KR-20 for the knowledge questionnaire was 0.7003.

Table 1 Descriptive information of the participants for Study 1.

	Sample
Size	Mean
Std.err	Std dev	Mean age
std.err	M/F ratio	
Study 1: Knowledge					
Control	752	0.52 ± 0.01	0.18	11.63 ± 0.08	1.08	
Experimental	797	0.75 ± 0.01	0.15	11.87 ± 0.08	0.88	

For the overall analysis of the questionnaire, a score (questionnaire score) was computed based on the proportion of correct responses. The number of correct responses divided by the total number of responses (correct and non-correct responses) was calculated. Additionally, and following the questionnaire score method, two global scores were extracted, one for the questions related to the knowledge of the species category (questions 2, 5, 6 and 7) and one for the questions related to the knowledge of conservation and welfare category (questions 1, 3, 4 and 8). In a further analysis, these two categories were compared globally, by gender and EEA.

To know the effects of the predictors (EEA, gender, and grade) in the questionnaire score, we used a linear model (Baayen, 2008). Standard linear Models were run in R version 1.2.533 (R Core Team, 2020) by lmer function (Bates et al., 2014). We generated various models and selected the model in which the predictors offered the most parsimonious combination through the corrected Akaike information criterion (AICc) (‘aictab’ function) (Burnham & Anderson, 2002). We checked whether our models fit the criteria of normal distribution and homogeneous residuals by visual explorations of histograms and qqplot of the residuals as well as residuals plotted against fitted values. We checked the significance of the predictors at the global level by contrasting the full model and the null model, excluding all predictors (Dobson & Barnett, 2008; R Core Team, 2020). Finally, we used a Chi-squared test to evaluate the effect of the EEA for each one of the items of the questionnaires. In this way we compared the total volume of correct responses of each of the questions between control and experimental groups. Descriptive statistics (mean, median, standard deviation, maximum, minimum) for each item question, (control and experimental groups), were also calculated. In addition, the demographic variable age in the sample of children from experimental group and control group were compared using a t-test.

Knowledge assessment results

A total of 1,549 responses were recorded during the data collection campaign. Of these, 752 (48.54%) corresponded to the control group and 797 (51.46%) corresponded to the experimental group. The largest volume of participants came from secondary schools, aged 12 to 16 years (n = 890), and from primary schools (n = 609), aged 8 to 12 years. High school students, ranging in age from 17 to 18 years (n = 50), participated less frequently (Table 1). In addition, the frequency of male students and female students was similar, 49% and 51% respectively.

Overall impact of the EEA on knowledge assessment

The best fit model (AIC = −1197.88) included the predictors (1) EEA, (2) grade, (3) and gender, (Table 2). The best fit model compared to the null model was significantly better at predicting the score of correct responses (χ2 = 203.08, df = 4, P < 0.001).

Table 2 Model selection for Study 1.

It shows the five most highly supported models developed to assess the impact of environmental education on knowledge. Models are ranked by ΔAICc.

Model	EEA	GRADE	SEX	AICc	ΔAICc	Weight	
Full Model	X	X	X	−1197.88	0	1	
Model 2	X			−1180.12	17.76	0	
Model 1		X	X	−1178.60	19.28	0	
Null Model				−551.51	646.38	0	
Notes.

ΔAICc, Akaike’s information criteria.

EEA, environmental education activities.

The generated model showed a significant effect of the predictor EEA in the proportion of correct responses (F = 788.520; df = 1; p < 0.010). In this sense, participants produced a significantly higher volume of correct responses in the experimental groups (Fig. 2). The model revealed significant differences in the grade predictor (F = 11.71; df = 2; p < 0.001). According to the post-hoc, primary school students (mean = 0.613; sd = 0.006) obtained a significantly lower number of correct responses than did secondary school students (mean = 0.654; sd = 0.005), (t = −4.738; df = 1; p < 0.000). On the other hand, gender was a non-significant predictor variable. (F = 0.7721; df = 1; p < 0.533) (See Table S1).

Figure 2 Proportion of correct responses for the control and experimental groups for the knowledge assessment.

Impact of EEA on question responses

In general, the students chose more correct responses in the experimental group. Only in question 7 the experimental group showed a decrease in the percentage of correct responses (Fig. 3). Educational activities showed the greatest impact in questions 6 and 8. For question 6, the volume of correct responses in the experimental group was 50% higher than in the control group. (χ2 = 296,242; p < 0.010). This difference was 44% for questions 8 (χ2 = 300.456; p < 0.010) (Table 3).

Figure 3 Proportion of correct responses for the control (CO) and experimental (EX) groups for the knowledge assessment for each question.

Table 3 Value of the contrast test for each of the questions between the control (CO) and the experimental (EX) groups related to knowledge assessment.

Questions	N	M_CO ±SD	M_EX ±SD	χ 2x	P	
1. If you can have a chimpanzee at home since being a baby, do you think it could be a good pet?	1,538	0,47 ± 0,50	0,86 ± 0,34	273,037	0.000	
2. The chimpanzee is A solitary/social animal/ He lives in groups/on his own	1,531	0,92 ± 0,27	0,97 ± 0,15	19,636	0.000	
3. Do you think chimpanzees are good for TV commercials?	1,538	0,80 ± 0,40	0,95 ± 0,20	86,724	0.000	
4. Does training animals like chimpanzees to participate in movies or commercials hurt them?	1,533	0,56 ± 0,50	0,75 ± 0,43	56,033	0.000	
5. Chimpanzees are endangered/ vulnerable/not threatened	1,529	0,45 ± 0,50	0,70 ± 0,46	99,393	0.000	
6. How many years a chimpanzee can live in captivity?	1,518	0,37 ± 0,48	0,87 ± 0,40	296,242	0.000	
7. How much can an adult male chimpanzee weigh?	1,527	0,31 ± 0,46	0,25 ± 0,43	8,599	0.003	
8. A primate rescue center like the Mona sanctuary:
(a) Rescues and socializes primates that come from circuses, TV commercials and the pet trade.
(b) Heals the chimpanzees and then takes them back to their habitat.
(c) Heals primates injured in the jungle.
(d) All of these are correct	1,530	0,31 ± 0,46	0,75 ± 0,43	300,456	0.000	
Notes.

CO Control

EX Experimental

Impact of EEA on knowledge by categories

If we cluster the analysis of the questions according to the categories: knowledge of the species and knowledge of primate welfare and conservation, the LMM shows that all participants obtained more correct responses on the questions related to the knowledge of welfare and conservation than on those related to knowledge of species (Category; F = 135.3642; df = 1; p < 0.010). In terms of gender, female students scored more correct responses on items related to the knowledge of welfare and conservation compared to male students (Category × gender; F = 11.007; df = 1; p < 0.010) (Fig. 4). No significant gender differences were found for the cluster of questions related to knowledge of the species (See Table S2).

Study 1. Knowledge assessment discussion

Overall, in this study, children in the experimental group responded more correct responses than did those in the control group, which suggests that the content provided in the educator-led visits, as well as the experience of engaging in a visit of close proximity to primates, resulted in children increasing their knowledge about primate species, their welfare and conservation. Question number 7, “how much an adult male chimpanzee can weigh?”, was the only question that had less correct responses in the experimental group compared to the control group. There could be various explanations for this result. First, the response options were not very clear as the thresholds for chimpanzee weights were very similar, thus the question could have been difficult. Another explanation could be that when educators talk about the dimorphism of the species, they talk about the weight of males and females, and perhaps participants do not retain the information because either it is confusing, or they do not consider it relevant. Given these results, this question could be adapted for future questionnaires, because the most important fact for children to retain is that chimpanzees are heavy wild animals which does not make them good pets.

Figure 4 The linear prediction of the proportion of pro-conservation responses according to gender, for each category.

In terms of grade level, secondary school children chose more correct answers on the knowledge questionnaires than primary school children. These results agree with studies carried out by Borchers et al. (2013), in which differences were observed in terms of academic training in students in the fifth and sixth grades. Burnett et al. (2016) also found that children in higher grades had better scores. These results could be related to the fact that high school students have more learning experiences (Borchers et al., 2013) and may be more familiar with environmental education concepts and terms than participants in earlier grades. High school students likely gain more knowledge about educational action than younger age groups (Lawson, 1983). Another explanation could be related to the type of questionnaire used. For instance, unlike the attitude questionnaire (visual type), the knowledge questionnaire utilized written questions and answers. Thus, none of the questions and answers were adapted to grade level. All participants, regardless of grade level and background, received the same questionnaire. As educators did not help children when answering the questions, it is possible that students in lower grade levels had more difficulty understanding some of the terms and questions in the questionnaire than higher grade level students.

Questions included in this knowledge assessment fell into two categories: knowledge of the species (questions 2,5,6 and 7) and welfare and conservation of chimpanzees (questions 1,3,4 and 8). When we analyzed the questions according to the categories, we observed important differences. Participants obtained better scores in the “conservation category” than in the “knowledge of the species category”. One explanation could be related to the type of visit and the predisposition of the educators, who might be more focused on conservation than on general knowledge of the species. Similarly, it is possible that participants had more general learning experience related to conservation than related to chimpanzee-specific issues, and therefore performed better in the conservation category. Additionally, this study showed an interaction between gender and the welfare and conservation category. Female students scored better than male students in this category, although no significant gender differences were found for the cluster of questions related to knowledge of the species. This finding is consistent with other studies (Bogner & Wiseman, 2004; Wiseman, Wilson & Bogner, 2012), which suggest that female students show stronger pro-conservation attitudes than male students.

Study 2: Attitude assessment

Attitude assessment methods

In this study, we assessed participants’ conservation attitudes toward primate welfare and conservation. We used a questionnaire to assess participants’ choice for pro-conservation messages over non-conservation messages. The questionnaire consisted of eight questions, each with two possible responses. Photos rather than text were used to represent each answer: one pro-conservation option and one non-conservation option (Fig. 5).

Figure 5 Attitude questionnaire.

The list of questions in the attitude questionnaire.

This attitude assessment method was based on a questionnaire developed by Bowie et al. (2020) in order to evaluate conservation attitudes among students participating in the education program at a primate sanctuary in the Democratic Republic of Congo. Some of the survey questions referred to the choice between a bonobo as a pet and a bonobo in the wild. The novelty of the questionnaire was that questions asked participants to choose ideas to design publicity to attract more visitors to the sanctuary hoping to have more unconscious ideologies when answering the questions. In conventional attitude assessments, participants often respond with answers they think are correct instead of providing answers that accurately convey their actual beliefs (Falk et al., 2007). For our study, we eliminated 4 out of the 12 original questions to concentrate the participants attention on chimpanzees that have been poached, trafficked, and/or are living in inadequate situations in Europe. Specifically, questions focused on topics like the presence of soldiers in the streets, bonobos being sold as bushmeat and life conditions in African countries. We included a preliminary section on demographic details including gender, age, grade, and school name (Table 4). A total of 1656 schoolchildren participate in this study. In terms of educational levels, 48% were in primary school, 43% were in secondary school and 9% were in high school.

Table 4 Descriptive information of the participants for Study 2.

	Sample
Size	Mean
Std.err	Std dev	Mean age
std.err	M/F ratio	
Study 2: Attitude					
Control	822	0.72 ± 0.01	0.18	11.61 ± 0.09	0.92	
Experimental	834	0.86 ± 0.01	0.15	11.59 ± 0.09	0.82	

For the attitude questionnaire, we proceeded using the same methodology as in Bowie et al. (2020) as we have used a similar questionnaire as the one used by the author. We calculated the total number of pro-conservation messages divided by the total number of messages (pro-conservation and non-conservation). In all the cases, continuous scores from 0 to 1 were obtained for each questionnaire and participant. To know the effects of the predictors (EEA, gender, and grade) on the questionnaire score, we used the same models as those used in the knowledge study. In addition, the demographic variable age in the sample of children from experimental group and control group were compared using a t-test. As in the knowledge questionnaire, we measured the internal consistency of the attitude questionnaire with a Kuder-Richardson Formula 20 (KR-20) (Kuder-Richardson Formula, 2014). The KR-20 for the attitude questionnaire was 0.711.

Attitude assessment results

A total of 1,656 responses were recorded during the data collection. Of these, 822 (49.63%) corresponded to the control group and 834 (50.36%) corresponded to the experimental group. The largest volume of participants came from primary schools, aged 8 to 12 years (n = 798) and secondary schools, aged 12 to 16 years (n = 712). High school students aged 17 to 18 years (n = 146) participated less frequently (Table 4). Additionally, 53.51% of the responses were from female students (n = 886) and 46.49% were from male students (n = 770). There were no significant differences (t-test = 0.1731, df = 1654, p-value = 0.8626) in the age variable of the sampled children exposed to or not exposed to environmental education.

Overall impact of the EEA on school children’s attitudes

The best fit model (AIC = −1308.61) included the predictors (1) EEA, (2) and gender, (Table 5). The best fit model compared to the null model was significantly better at predicting the score of correct responses (χ2 = 191.93, df = −2, P < 0.001).

Table 5 Model selection for the attitude questionnaire.

It shows the five most highly supported models developed to assess the impact of environmental education on attitudes. Models are ranked by ΔAICc.

Model	EEA	GRADE	SEX	AICc	ΔAICc	Weight	
Model 1	X		X	−1308.61	0	0.85	
Full Model	X	X	X	−1305.06	3.85	0.15	
Model 2		X		−1275.03	33.58	0	
Null Model				−966.93	341.68	0	
Notes.

ΔAICc Akaike’s information criteria

EEA environmental education

The model revealed a statistically significant effect of the EEA predictor (EEA; F = 347.2837; df = 1; p < 0.001) and gender predictor (gender; F = 34.2497; df = 1; p < 0.001) on the proportion of pro-conservation responses. Our results showed that pro-conservation attitudes were influenced by EEA and by gender. Participants produced a significantly greater proportion of pro-conservation responses in the experimental (Mean = 0.86, std. err = 0.006) group than in the control group (Mean = 0.72, std. err = 0.006). In terms of gender, female students (Mean = 0.815, std. err = 0.005) showed a greater proportion of pro-conservation responses than male students (Mean = 0.767, std. err = 0.006).

Impact of EEA on attitude questions responses

Figure 6 shows the percentages of choices of pro-conservation messages for each of the questions. Except for question 4, in which participants showed a similar percentage of responses between the control and the experimental group, participants had a significantly higher proportion of pro-conservation responses in the experimental group. Mona’s EEA produced a significant positive change in pro-conservation messages in seven of the eight questions surveyed. EEA activities showed the greatest impact in questions 6 and 7. For question 6, the proportion of pro-conservation messages increased by 33% in the experimental group. Responses to question 7 the pro-conservation messages were 38% higher in the experimental group than in the control groups (Table 6; Fig. 6).

Figure 6 Proportion of pro-conservation responses for the control and experimental groups on attitude assessment.

Table 6 Value of the contrast test for each of the questions between control (CO) and experimental groups (EX) for the attitude questionnaire.

Question	N	M_Control (CO)
(SD)	M_Experimental(EX)
(SD)	X2	P	
1. Which group do you think chimpanzees belong to?	1,637	0,89 ± 0,32	0,93 ± 0,25	10,910	0.001	
2. Which of these photos do you prefer to see in a Mona sanctuary ad?	1,642	0,70 ± 0,46	0,85 ± 0,36	53,736	0.000	
3. Which group do you think chimpanzees belong to?	1,644	0,62 ± 0,48	0,75	31,972	0.000	
4. Which photo best shows the value of the forest?	1,640	0,97 ± 0,16	0,98 ± 0,15	0,272	0.602	
5. Which group do you think chimpanzees belong to?	1,639	0,83 ± 0,38	0,89	15,059	0.000	
6. Which of these photos do you prefer to see in a Mona sanctuary ad?	1,644	0,54 ± 0,50	0,87 ± 0,33	208,268	0.000	
7. Which of these two situations do you prefer to be in?	1,627	0,40 ± 0,49	0,78 ± 0,41	239,383	0.000	
8. How do you like to see this chimpanzee?	1,644	0,83 ± 0,37	0,94 ± 0,24	45,418	0.000	

Study 2. Attitude assessment discussion

Overall, participants in the experimental group were significantly more likely to choose the pro-conservation responses than in the control group. These findings suggest that the MonaEduca program seems to be well-oriented in promoting pro-conservation behavior and positive attitudes in children, thus we believe that it is fulfilling a very clear objective as a precursor to these changes. Although the literacy levels among our participants were similar, we believed that using pictures as choice options instead of written response options would facilitate the predisposition to answer the questionnaires (Bowie et al., 2020). On another level, providing electronic devices such as tablets to answer the questionnaires encourages children to participate.

One of the key messages of the MonaEduca program is that primates face threats from the international pet trade. International routes of primate trafficking indicate that Europe is one of the main markets for great apes (Stiles et al., 2013) thus the educator-led visit emphasizes that primates are not good pets. How primates are featured in social media can push people to have them as pets (Ross, Vreeman & Lonsdorf, 2011; Aldrich, 2018). Thus, for the MonaEduca it is very important to convey an anti-pet trade message to children, who are exposed to content on social networks that do not place value on primates or primate conservation. Thus question 8 asks “How would you like to see this chimpanzee, as a pet or in the wild?” This was the question that received more pro-conservation responses, as 93% of children in the experimental group answered that they would prefer to see a chimpanzee in the wild.

In this study, we found that female students were more likely to choose pro-conservation responses than male students. Females seem to show stronger moral attitudes than male students (Eagles & Demare, 1999; Bogner & Wiseman, 2006). As defined by Kellert (1982) a moral attitude refers to one’s concern for right and wrong when it comes to our relationship with animals. This result does not have a clear explanation. Some studies suggest that formal education’s social and cultural environments may predispose males to a utilitarian use of the planet and females to a more protective attitude towards nature and animal conservation due to gender socialization (Zelezny, Chua & Aldrich, 2000; Arnocky & Stroink, 2011; Xiao & McCright, 2012). These studies suggest that females usually report stronger ecocentric environmental attitudes than males (Zelezny, Chua & Aldrich, 2000), show more respect toward animals than men, and are significantly less anthropocentric and more compassionate (Kaliský & Kaliská, 2022). However, there might be other explanations related to specific aspects of the environmental activity, such as the activity itself or the gender of the educator. Shutts and colleagues (2010) show evidence that children’s learning is influenced by gender. The authors observed that human infants tend to retain information more effectively if the educator or informant is of the same gender. As such, according to Shutts and colleagues (2010), one possible explanation for females scoring more pro-conservation responses might be that most educators from the MonaEduca team were women (4 versus 1).

General Discussion

Our study is the first assessment of an environmental education program in Europe carried out at a primate sanctuary. In this study, we evaluated the MonaEduca program, which focuses on stimulating changes in attitudes and increasing knowledge of primate conservation among primary, secondary, and high school children aged 8 through 18. According to our hypothesis, the results of this study demonstrate that an educational program carried out in an informal setting such as a primate sanctuary has a positive effect on children’s attitudes toward primate welfare and conservation and can support short-term knowledge acquisition. As an outdoor activity, the visit to a primate sanctuary can have a major impact on children’s emotions as they have an up-close encounter with the animals as happening in zoos (Hacker & Miller, 2016). Learning firsthand the reasons why the animals are housed in the sanctuary, and their life history and behavior (Prokop, Tuncer & Kvasničák, 2007) supports the formation of an emotional bond that is related to the will to protect the animals and their natural environment (Zhang, Goodale & Chen, 2014). Our results are aligned with those of other studies carried out in animal reserves, national parks, and biosphere reserves (Kuhar et al., 2007; Borchers et al., 2013; Rakotomamonjy et al., 2015; Burnett et al., 2016; Grúňová et al., 2017). These results are also supported by studies carried out in primate sanctuaries such as the Lola Ya Bonobo sanctuary in the Republic of Congo (André et al., 2008) and other sanctuaries in Central Africa (Kuhar et al., 2012), as well as studies carried out at zoos (Lukas & Ross, 2005; Moss & Esson, 2010; Jensen, 2014; Moss, Jensen & Gusset, 2014; Chalmin-Pui & Perkins, 2017; Moss, Jensen & Gusset, 2017a; Moss, Jensen & Gusset, 2017b; Spooner et al., 2019; Collins et al., 2020).

Implications for the development of educational programs in animal sanctuaries

The results of this study suggest that the work carried out in education conservation programs with children in a sanctuary is an important aspect in improving knowledge and attitudes toward charismatic species such as primates. Overall, all children from the 83 schools that participated in this study showed improvement in their attitude and knowledge after completing the program. Conservation education programs are very rarely integrated into the national education curriculum. Thus, school children engage in this type of outdoor program at a sanctuary as an extracurricular activity. These programs carried out outside of school, aim to reinforce the curricular program but also to help children engage in new activities provided by the opportunity to visit a sanctuary near their school. These types of positive experiences with nature and animals during childhood promote children’s commitment to protecting the environment when they become adults (Wells & Lekies, 2006; Chawla & Cushing, 2007). The work carried out at primate sanctuaries like Mona can help to engage emotions in children when they are exposed to primates that are in a rehabilitation process. These emotions help to connect participants with the animals and improve children’s attitudes toward primate welfare and conservation (Clayton, Fraser & Saunders, 2009). Emotional connections made during environmental educational programs in young people are the main triggers of the measured outcomes (Stern, Powell & Hill, 2014). Given the unfortunate disconnect between children and wildlife, it can be important to allow children to experience nature as these experiences have been shown to promote biodiversity conservation (Schuttler et al., 2019). Additionally, the primary use of data and all knowledge generated by this study will be used directly to improve the current MonaEduca Activities. This will in turn allow Mona to positively impact children’s emotions and empathy towards primates.

Limitations and Future Research

The main limitation of this study was that the tool we used measured only the immediate impact of the educational activity. Although the short-term effects of the activity are encouraging, we cannot know how long participants will sustain their new knowledge. What we know is that attitude changes in children can be transmitted to adults (Rakotomamonjy et al., 2015). The emotion children show when engaging with the stories of rescued chimpanzees is likely transmitted to their families as many families decide to visit the sanctuary after a school visit from their child (Feliu, pers. comm., 2020). The schools participating in the study came from different urban and rural areas, with very different levels of family education and socioeconomic status. Personal experiences, outdoor activities, and families influence the attitudes of children toward nature (Eagles & Demare, 1999). Since we did not have access to these sources of information in our surveys, we cannot know how these variables influenced our results. Our survey instrument did not contain any personal questions that measured student involvement in other environmental activities outside school that influence participants’ environmental attitudes.

As there are few studies of conservation education with school children in sanctuaries, it is difficult to compare the results obtained from Mona’s educational program with the programs of other sanctuaries. In the absence of such studies, we encourage primate sanctuaries to promote empirical studies of attitudes and knowledge of primate welfare and conservation and conduct systematical evaluations to strengthen their educational activities. It is important to assess if these programs work, but also why and how they work (Stern, Powell & Hill, 2014). Despite these caveats, we believe that the information presented in this study is very valuable. It is one of the first studies that shows the critical importance of the educational role of primate sanctuaries.

Supplemental Information

Supplemental Information 1 MonaEduca Activities

Click here for additional data file.

Supplemental Information 2 Children’s attitudes data

Click here for additional data file.

Supplemental Information 3 Children’s knowledge data

Click here for additional data file.

Supplemental Information 4 Results of the linear model for the questionnaires

A) Results of the linear model for general impact of Knowledge. B) Results of the linear mixed model for general impact of Knowledge for category. C) Results of the linear model for general impact of attitudes.

Click here for additional data file.

Supplemental Information 5 Knowledge questionnaire in Catalan

Click here for additional data file.

Supplemental Information 6 Attitude questionnaire in Catalan

Click here for additional data file.

Supplemental Information 7 Knowledge questionnaire in English

Click here for additional data file.

Supplemental Information 8 Attitude questionnaire in English

Click here for additional data file.

Thanks to Emily Felt for the English proofreading, to Martí Masip for helping with the bibliography to the Collaboration Agreement between Fundació Mona and Instituto de Investigaciones Biológicas of the Universidad Veracruzana, and to the MonaEduca team for doing such a great job and for organizing the children for the data collection. Thanks to all the teachers and children that participated in the survey.

Additional Information and Declarations

Competing Interests

Author Contributions

Human Ethics

Data Availability

The authors declare there are no competing interests.

Olga Feliu conceived and designed the experiments, performed the experiments, authored or reviewed drafts of the article, and approved the final draft.

Arturo González-Zamora conceived and designed the experiments, authored or reviewed drafts of the article, review, editing and supervision, and approved the final draft.

David Riba conceived and designed the experiments, performed the experiments, analyzed the data, prepared figures and/or tables, authored or reviewed drafts of the article, and approved the final draft.

Teresa Sauquet conceived and designed the experiments, authored or reviewed drafts of the article, and approved the final draft.

Sònia Sánchez-López conceived and designed the experiments, authored or reviewed drafts of the article, review, editing and supervision, and approved the final draft.

Carmen Maté conceived and designed the experiments, authored or reviewed drafts of the article, review, editing and supervision, and approved the final draft.

The following information was supplied relating to ethical approvals (i.e., approving body and any reference numbers):

Ethical approval was granted by the Bioethics Committee of the University of Barcelona.

The following information was supplied regarding data availability:

The raw data is available in the Supplemental Files.

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
