# Peer review of "The impact of sanctuary visits on children’s knowledge and attitudes toward primate welfare and conservation"

_PeerJ, doi:10.7717/peerj.15074_

## Round 0.1 · original submission · Major Revisions

I have been extremely fortunate to receive three very detailed and conscientious reviews from experts. All three reviewers recognized the strengths of your study including the large sample size and importance of the question. For this reason, I am pleased to invite you to engage in a major revision of your paper. However, each reviewer also identifies the need to reframe and reorganize the study presentation. Each has made numerous helpful suggestions for improving the clarity and accuracy of the writing. I will not reiterate these points since the reviewers are quite clear. I will also not engage in my own detailed copy-editing at this point given the number of changes required. However, I do have a few additional comments of my own. Whereas the reviewers appreciated the between subjects design of the study, I feel that there needs to be more justification for not using a within-subjects design to directly test whether the experience had an effect on attitudes. If the goal was to compare two groups that were equally engaged by the desire to visit a sanctuary while controlling for exposure by testing half of them beforehand, this should be presented as part of the rationale for the design. You should be careful to address potential confounds given that these were two different groups of participants and that the "post" group completed the surveys at the end of the day with potentially greater demand characteristics following the visit compared to before. These limitations need to be addressed in the discussion. Of course, the issue of demand characteristics would arise in within-subjects design as well, but it is a bit odd to conduct a pre-post manipulation across different participants. I agree with Reviewer 3 that calling it a pre/post measure is a bit confusing. Perhaps call it control/experimental groups?

Because you randomly assigned different questionnaires to different subjects, it would make more sense to present this as two separate studies given that the questionnaires asked different questions and tapped into different outcomes, and involved different participants.

I agree with Reviewer 3 that it would be helpful to know more about the questionnaires in the main text. Your description should follow typical Method/Materials sections for survey research indicating how many questions per survey, any subscales, the response scale, the scoring and sample items, as well as information on validity and reliability. Please indicate exactly which questions you removed from the Bowie survey.

Did you engage in any "data cleaning" to eliminate random responses or incomplete data?

Please report how many males and females were in each of the four samples (two surveys, two testing times). Please report mean age with SD for each sample as well. Please report confidence intervals and effect sizes where appropriate. Be more transparent about the models. How were sex and age coded? It seems like you used grade levels by school rather than age. Was this dummy coded? Into how many categories? Be clear about this and justify it. Allowing age to vary continuously rather than collapsing participants by grade level might be more nuanced.

The heading "Impact of questionnaire items on knowledge" doesn't make sense to me. Items can't cause knowledge. Also, you cannot talk about decreases or increases across the questionnaires (e.g., lines 287, 390) because that implies a time related change but it was completely different participants responding at the two time points.

Be consistent in your use of sex or gender - don't switch between these terms.

Make sure the labels in the figures are in English (e.g., x axis Fig. 3). The quality of the figures need to be improved.

Don't plot the interaction of sex by condition in Figure 5 because you did not find a significant interaction. You can simply state the means and SDs to explain the main effects.

I look forward to receiving your revision.

Jennifer Vonk

Reviewer 1 ·

Basic reporting

See below for details, but overall writing needs improvement. Both in English writing but also organization and presentation.

Experimental design

Experimental design and execution was sound. I am excited to see this work being completed. I do have questions related to statistics but nothing insurmountable. See details below.

Validity of the findings

No comment.

Additional comments

Introduction

Introduction as a whole is a bit disorganized and I struggled with following the broader points.
- The opening paragraph(s?; spacing unclear, line 50 and 57) are a bit scattered in focus. In addition, the content itself is a bit confusing. Each paragraph should have a relatively clear point or points that conclude to a specific message. What that is here is unclear.
- Paragraph 63: What is point/focus here? It goes from EE is important, primate sanctuaries are good for EE (but already set that up in earlier paragraphs), childhood is important for EE, child EE at sanctuaries is a priority. This section seemed rush and a lot of half explained content included. I would recommend to the authors to focus on what are the most important points for setting up the context of this study and focus on describing the literature on that and how it applies to their study.
- One part that stood out was that a lot of effort was made to set up primate sanctuaries as a good place for EE, then a large section was devoted to zoo EE. If emphasis here is on sanctuaries, focus on that work. Citations of zoo work is relevant in that it provides context, but I feel you could make that point in a couple sentences rather than several paragraphs, keeping emphasis on your study context and saving space to discuss/set up more relevant context.
- Line 114: The authors go back to sanctuary education.

Line 53-54: “today sanctuaries are essential in providing environmental education (Hansen et al. 2016; Kuhar et al. 2012).” Is saying “essential” an appropriate reflection of these citations? Essential to what or to whom? These two citations highlight potential values of such programs in two contexts (US and Africa) but if labeled as “essential” then the statement should be more specific to as what they are essential to and how these cited studies highlight that. It may be more accurate to state that sanctuaries “contribute to” or are “ideal environments for” environmental education.

Line 54: “that helps end the trade of species”. Is this the context you intended for the above sentence? Do you have a citation to support this? I value sanctuaries and environmental programs but I am not sure there has been a clear or even less than clear demonstration that sanctuary education programs have impacted the actual trade of primates. This needs appropriate citation or a reworking of this broader sentence to properly put sanctuary education programs in context.

Line 59: “Many of the animals traded out of their habitat end up in sanctuaries in non-range countries…” Is this true? A citation here would be beneficial. I would say many in-range sanctuaries work this way, but my familiarity with non-range would say they came from other places (biomedical, entertainment, etc.) not to say some of those weren’t traded illegally, but the phrasing of this sentence is specific to a point that it needs a citation or reworking to put the introduction in the proper context.

Methods

Line 167: Can detailed information related to points 1 and 3 be shared as supplementary material? This would provide more context to readers and make a blue print for replication available to interested parties.

Line 174: This section had variable sentences that should be reworked for clarity and/or flow and several grammatical errors. For example but not limited to:
- Line 175 – “who participated in the study, 1526 Boys..” the comma going into the numeric values reads oddly (maybe better in () or a table). Capitalization of sexes is incorrect.
- Line 182 – “participating in the MondaEduca activities was previously asked permission to participate in the study”
- Line 184 – “University of Barcelona approved to conduct of this research with participants from school participants.”

Line 222: The specification of Lola Ya bonobo seems odd here, would be more appropriate to say “at a primate sanctuary in Democratic Republic of Congo.

Line 223: Expanding on above comment these two sentences are not needed and seem to confuse the reader. Should be removed.

Line 235: Authors should clarify their use of the term “global” as it is used here and throughout and is not obvious. Using a term more specific to their aims, rather than a generic statistical label, may improve clarity.

Line 245: “fixed predictors EEA” – it took me a while to realize you mean Pre/Post here. I would recommend labeling or wording this better to highlight this point.

Line 234: These comments relate to the statistical analysis presented in this section.
- The authors state that they used a form of model selection. The authors need to then provide a table that shows each individual model tested and how they compared to each other model (AIC score and change in score).
- The authors state that they compared a full model to null model, this statistical output needs to be shared.
- Forstmeier and Schielzeth, 2010 should be 2011 (as it is in reference section).
- The authors here have used the Forstmeier citation incorrectly. This paper argues against model building/selection and suggests authors should present a full model. As stated in the abstract of the article “We favour the presentation of full models, since they best reflect the range of predictors investigated and ensure a balanced representation also of non-significant results.” This citation should be removed in favor of one that supports their model selection approach. Or more ideally, model building/selection here is removed entirely and the authors present the results from the full model. It seems the authors may have done this as they mention the full/null comparison but not clear if the full is the selected for model or true full. I prefer the authors only present a full model and abandon the model selection approach and there are several other strong arguments in support of this (see for example Mundry & Nunn, 2009 and Mundry 2012, plus the Forstmeier citation included), however, I recognize model building/selection is common and accepted and don’t wish to argue against it entirely, but I would suggest the authors here consider it strongly as it is less prone to Type 1 error, requires less work in conducting (so shouldn’t be a challenge to redo here) and provides clearer and more interpretable methodology and results to readers.

Results

Line 271: Here but elsewhere you report P = 0.00, I imagine this is a rounding consideration but I was taught to not round down to 0.00 and in reference to P values to make it always P < 0.01 or 0.001 depending on how you are rounding your decimal points. Since F value is to three decimal points the P value should also be so P < 0.001

Line 276: “as this variable had no impact on the model.” This is confusing as by being included in the model this variable had an impact on the model, it just was not a significant predictor of the outcome variable. This phrasing is used in several placed. Presentation of this is best described as being “non-significant predictor variable.”

Line 273: “According to the post-hoc and as expected” wording like this that provides context/interpretation is used in several places in results (for example also on line 288). Authors should refrain from interpreting their results in the results section and focus on just reporting the quantification. Context should be limited to understanding what the quantification is but not its implications to the broader study.

Line 306: “of species” should be added after knowledge to clarify which questions you are describing.

Line 342: “The figure shows…” Which figure? Please specify. Also, when describing the results one should not reference the figure as the main point of emphasis but rather describe the data and then reference the figure/table to help clarify or reinforce the data/analysis description.

Figures: Figure labels seem unedited from the output of the software using underscores and abbreviated titles and auto generated labels. Possible edits to increase readability/presentation.

Discussion

Opening Paragraph: Here the authors highlight a series of sanctuary education programs but none were introduced in the introduction. The citations in the discussions should mostly reflect those in the introduction, not including these in intro affects context set up and makes interpretation in discussion more challenging since the authors did not introduce these points prior to their discussion/comparison with their findings.

Line 428-431: I would expect more discussion of this finding as this point is highlighted in study title. What else does this finding suggest?

Line 462: “Females are less prone to utilize animals and more prone to preserve them… due to gender socialization” do the citations at the end of the sentence relate to this point or the second half of sentence. The wording here is also vague, what does “utilize” mean and how are you interpreting it? What does “due to gender socialization mean, how are you using this here and how is it related to your cited studies? This conclusion and citations are generally unclear. This should be discussed in more specific detail, it is not clear what is being said here.

·

Basic reporting

This study investigated the changes in knowledge and attitudes related to conservation in schoolchildren who visited a primate sanctuary in Catalunya, Spain. The authors found that the children demonstrated increases in knowledge and positive attitudes towards primate welfare and conservation after their visit. They also found that female students demonstrated more knowledge gains about chimpanzee conservation than male students. This paper will require further editing for grammar and punctuation (comments listed below).

Experimental design

This study had a large sample size and addressed a gap in the literature. Although there are a number of studies of this type that have been conducted after students have visited national parks in primate range countries, very few studies have examined whether visits to sanctuaries have similar benefits. The methods are appropriate and the figures illustrate the important findings.

Validity of the findings

My main concern is the interpretation of the finding that female students scored higher on the chimpanzee conservation questions than the male students (lines 454-465). This section implies that the reason that female students scored higher on these questions is that girls/women are naturally more compassionate and caring than boys. I think this is a problematic explanation for why female students scored higher, and reinforces sexist stereotypes. I find the sentence in lines 461-461 to be particularly concerning: “Gender differences, towards animals are linked to a female’s caring morality (Koleva et al., 2014), heightened emotions and empathy (Hills, 1993).”

First, I do not think that doing better on the conservation questions necessarily indicates that female students “were more sensitive to primate conservation issues” (454-555) than male students, as there could be numerous other explanations for why female students scored higher. Maybe the female students paid more attention to the tour, for some reason. Maybe female students listen more closely to female educators than boys do. Maybe the photos chosen for the questionnaire were more appealing to one sex or the other. I don’t know if any of these are true, but my point is that there are other possible reasons for these findings besides innate differences between the sexes that other research has shown do not exist.

Secondly, many of the citations listed are old (10-20 years or more), and when I looked into them, I do not find these particular citations to be convincing. Many have tenuous findings about the differences between women and men, or refer specifically to the way women are socialized in regards to social responsibilities. If you want to argue that female students are socialized differently than male students in Spain, and that is why female students scored higher on the conservation section, you will need to strengthen this discussion and present more evidence and support from the citations than is currently present.

Also, I recommend using the terms “female students” and “male students” instead of “girls” and “boys” throughout. Please be consistent when talking about adults as well; use either “females” and “males” OR “women” and “men,” but not a mixture of both (see lines 458 and 460 for uses of “females” and “men”).

Additional comments

There are numerous grammar and punctuation mistakes in the manuscript which need to be addressed. Specific comments below. The lines refer to the line number:

54: After the citation, rewrite to: “… that can help end the trade of species.”
71: “emotional” should be “emotionally”
70-71: This sentence is missing a word: should be “When people observe primates, they can connect emotionally with them.”
76: Insert a comma between “primates” and “promoting”
80: I don’t understand what is mean by “behavioral intention”
81: Insert a space after the citation and before the word “Therefore”
82: “Boost” should be “boosts”
83-85: These sentences are redundant and need to be rewritten. Perhaps something like “Childhood is the stage of life in which people develop an awareness of nature. Allowing children to interact with nature can increase their appreciation of the natural world.” This is still a bit redundant, but something along those lines.
93: “assess” is misspelled
106: Delete the extra comma after the word “youngsters”
125: Delete the underline under the period of “André et al.”
126: Delete the word “more”
127: Add a period at the end of the sentence
129: “Increases” should be “increased” In general, this whole article should be in the past tense.
130: “Fosters” should be “fostered”
131: The first “are” should be “were”
160: The grammar isn’t quite right here. Perhaps something like: “The curriculum targets all ages within primary and secondary schools [list age range here].”
161-162: This sentence should be two separate sentences, as it is two distinct ideas.
175: “boys” and “girls” should not be capitalized
176: “Range” should be “ranged”
193: Here, and everywhere else in the article, be consistent with your use of capitals for
“pre” and “post.” I suggest lower case.
193: “between subjects” should be plural
196: Add a space between 51 and percent
207: Delete the underscore before “Knowledge”
224: “bonobos” should be lowercase. Delete the extra period before “Some”.
225-227: The meaning of this sentence is unclear, please rewrite.
235: “score” should be lowercase
237: I think “no-conservation” is supposed to be “non-conservation”
242: “attitude” in “attitude questionnaire” is singular and not possessive, delete the “‘s”
243: Delete the space in “non-conservation”
245: “sex” should be lowercase
247: Remove the extra parenthesis. Double check this reference please (is the ‘y’ correct?)
252-253: “full” and “null” should be lowercase. Remove the bold formatting on “Finally”
266: Delete one repeat of the word “from”
270: “predictor” should be lowercase
271: Check the spaces before and after all equal signs. Be consistent.
281, 295: knowledge should not be capitalized
307, 308: Be consistent; use sex or gender
317: Should be “Attitude,” not “Attitudes”. Watch the plurals.
331: by “previous ones” do you mean pre-EEA questionnaires?
335, 338: Should be “attitude”, not “Attitudes”
342: Should be “choices”, not “choice”
352, 359: Should be “attitude”, not “attitudes”
389-390: no comma after sanctuary. “supports” should be “supported.”
393: “children” should be “children’s”
394: Add comma after “Overall”
396: “results” should be “resulted.”
398: I suggest using quotes around the question instead of parentheses. “experienced” should be “demonstrated”.
405: Change the word order: should be “heavy wild animals”
406: “obtained” is the wrong word. Maybe “chose”, or “selected”?
412: Delete the word “learning” in the middle of the sentence
420: “knowledge” should be lowercase
426-427: Change the verbs in the sentence to the past tense
441: Delete the period after “questionnaires.”
441-442: Add a comma after “level”, and change “encourage” to “encourages.”
446-448: This sentence does not make sense as written. Perhaps: “Thus, for MonaEduca it is very important to convey this message among children, who are exposed to content on social networks that does not place value on primates or primate conservation.”
449: Use either “can push”, or “pushes.” Delete the period before the citation
451: Delete the parentheses around the question
468: Delete “our hypothesis”
469: Should be “is an important aspect”, not “are”, because the subject of the sentence is “the work carried out”
471-475: This section is repetitive, and could be rewritten to be more concise.
481: Should be “in young people”, not “on”
483: Delete “with” before nature.
486: Delete “on” before “children’s emotions”
494: I would say, “is likely transmitted to their families”
499: Add “we” after “Since”
501: Delete “either”.
506: Should be “work”, not “works”
507: Delete “by”

·

Basic reporting

The following study examines children's attitude and knowledge following a visit to a primate sanctuary. The strengths of this study include a large sample of over 3200 children of various ages and a very thorough review of the existing literature. However, the manuscript could improve in the following ways.

1. Although the title is attractive, it is not the most representative of the study. Although girls gave better responses regarding welfare and conservation, all children showed gains in pro-conservation responses. This main finding is more compelling than the sex difference that the title implies.

2.The overall writing quality of the manuscript is good but there are several places where there it is unclear, or the English could be improved. Also, the Introduction and Discussion are quite lengthy and could benefit from additional editing.

3. ABSTRACT: Another strength of this study is the between subject design, presenting either the knowledge or attitude questionnaire before or after the visit. This is something that is not very clear (for example, line 34-36 in abstract reads as pre- post= test which would implies a within subject design). Also, in the Abstract (line 38) it states girls are more "sensitive". Not sure what is meant by this since you did not measure this sensitivity but knowledge and attitudes. Also, the last line of the abstract 39-40 is vague and not helpful. Please refer to your findings and specific suggestions for future research and specific suggestions for education programs.

4. INTRODUCTION: As mentioned earlier, the manuscript would benefit from additional editing for sentence structure and to condense some of the background literature. In addition, the last paragraph should introduce the purpose of the study, methods, and hypotheses in addition to the benefit of the study. A purpose is given but there are no formal hypotheses, and it isn't until much later in the Methods that the reader understands the design.

5. METHODS: Although the questions are found in supplementary material, it would be helpful for the reader to have an idea when reading the methods, the kind of knowledge and attitude questions ask. For example, line 212-214, you could state that questions 1-4 addressed general knowledge of the chimpanzee such as "do they make a good pet or does training for movies hurt them." Without looking at the questions, while reading the manuscript I interpreted knowledge to be facts about the species. Instead, what you are calling your "knowledge questionnaire" really contains several questions that are more ethics based. I would recommend you rename this survey or give a better explanation of what is in each of your surveys in the methods. Also, this also might explain why you saw difference between Q14 and Q 5-8.

6. RESULTS: All the figures could use some additional editing. Some are missing axis titles in English or are inappropriate for the data presented (Figure 3 for example). For Figures 2 and 6, because the questions are in the supplementary material, just listing Q 1-8 is meaningless to the reader. Either give the info in a method or perhaps a table with each question.

Experimental design

The sample size and between subject design are very good. It addressed a gap in the literature. There is missing information (stated above) that would help the reader understand what was done, how it was done, and the specific questions asked of the children.

Validity of the findings

Conclusions could use some additional editing. The authors give a very thorough review of previous research but how their study adds to what is out there is missing. Although there are not many studies done in sanctuaries, there is an abundance of zoo education studies that have examined short term changes in children's knowledge and attitudes. A comparison of findings here would be helpful.

---

## Round 0.2 · Major Revisions

Thank you for your continued work on the manuscript. You will see that one reviewer indicated the need for minor revisions while the other reviewer feels that not enough was done to improve the organization and clarity of the introduction in particular. Even the more positive reviewer had a number of remaining suggestions for clarity and improvement. Therefore, I would like to ask you to engage in more extensive revisions to the manuscript.

I have several of my own comments. Line numbers refer to lines in the reviewing PDF.

It should be clear from the abstract what the study design was. It has to be clear what your experimental variable is; that is, be clear that only one group was exposed to the program information. Be clear that different surveys were used for different participants. Did all children show gains, or only those in the experimental groups? The findings are not stated clearly. What is meant by “better” responses? More accurate or more detailed or just a greater number of correct responses? A reader needs to have a better idea of exactly what was done by reading the abstract. The abstract does not reflect the change to report two studies rather than one.

It sounds like the goal is to focus on discouraging the pet trade but then, on line 60 you state that one of the goals is to reinforce attitudes that promote primate conservation – please be explicit about how these goals align or focus solely on what the conservation education really focuses on. The line “Another goal...” starting on line 61 is not necessary. If your goal is to assess the efficacy of the program be clear what the goal is and exactly what is meant to change and how your measures are carefully selected to meet these goals.

I agree with Reviewer 1 that the structure of the introduction still seems unorganized and that the paragraphs do not have clear goals. Each paragraph should be fully developed with a clearly stated goal statement, a supporting body and a concluding statement. It seems the first four paragraphs could be combined into a single, more concise paragraph, about the important mission of primate sanctuaries and how their conservation programs attempt to serve this mission, and how your study will test its effectiveness.

Empathy is broader than just feeling the “welfare” of someone else (line 72).
By positive behaviors (line 73) do you mean prosocial behaviors?
Delete the . after “outcomes” on line 95.

Indicate early in the introduction why it is important to focus on children in particular.

You need to be more explicit about what you mean by knowledge and improved attitudes toward conservation.

Why include gender if you expected no gender differences? I think it would make sense to say gender was included because previous studies have shown gender differences in empathy toward animals and other related measures. Did you have hypotheses about grade level?

Can you say more about the learning-through-play activity?
Make it clear if all visitors experience all three aspects of the program.
The statement “in order to ensure high variability in the sample.” If all participants had completed both surveys, you would have the same variability in the overall study sample. Now you’ve just confounded it with the questionnaire presented. A better justification is needed for why each sample completed only one of the questionnaires. Again, this is less weird if you present the paper as two studies with differing goals – one to determine attitudes toward conservation and one to determine knowledge about primates.

The paper should be divided into Study One and Study Two before the Method begins b/c there should be a separate Method for each Study indicating the appropriate participant information for each questionnaire/study in separate subsections.

Please provide a reference for the Kuder-Richardson formula for internal consistency as this is not a well-known measure.

Line 235, this section is under “Study One” – the knowledge study so why are you referring to Pro-conservation responses here? It appears that you did a short of halfhearted reorganization of the paper rather than really carefully checking the consistency of the re-organization. Why are you talking about pro-conservation scores within the knowledge questionnaire here? This is all very confusing. Please include information about subscales and explain the type and content of questions and scoring in Materials subsections where each questionnaire is named and described fully. Lines 306-308 should be moved to Materials, for example. If these measures were created ad-hoc for the purpose of this study, that information also must be provided.

Line 241, what is meant by “by educational action?”

You mention excluding all fixed effects on line 250, but you haven’t told the reader what effects are fixed effects. What kind of linear model? A GLM or a standard linear regression?

I don’t understand the Chi square analysis mentioned in lines 251- 252. Do you have hypotheses related to specific questions?

It isn’t helpful to refer to Questions by Question # if the reader does not know what those questions refer to. Describe by content instead.

There are some underlined sections of text that need to be edited (removing the underline). Some of the fonts appear mismatched in the PDF as well.

More needs to be said about the implicit measure of attitudes in the introduction. Implicit measures have to be interpreted cautiously. There is a rich literature on the measurement of implicit attitudes and the challenges associated with them. Why did you decide to use this method here? There still needs to be more explanation for your design choices.

It is not clear how the scores are obtained for the implicit attitude measure.

To show that experimental and control groups are reasonably well matched, you should present descriptive statistics and frequency information for both groups in both studies showing the distribution by age/grade and gender. You may conduct t-tests to show that they do not differ in attributes (ideally) or to report if they do.

I agree with Reviewer 1 about just focusing on the effects from the full model rather than testing model fit.

Tables where the key data are presented should appear in the main text, not as supplemental figures or tables.

Figures 6 and 7 are not needed because you have only main effects. You can report the means for each group or gender.

Do not divide the General Discussion up by Study 1 and 2. Each Study should have its own mini-discussion and the general discussion should synthesize and present the data from both studies with reference to existing literature.
I agree with the point made by Reviewer 1 that you should avoid dictating how others should behave; e.g., “We must…” (line 539). Instead you might say something like “It can be important to allow children to experience nature as these experiences have been shown to promote interest in biodiversity conservation….(with ref)..” This interest is critical because…. Etc.

I agree with Reviewer 2 that, seeing the questions, makes it clear that not all of the items reflect attitudes or knowledge as implied by your labels for each questionnaire. Make it clearer what items are meant to assess. It might be helpful to indicate (maybe with different borders) which items are scored as “pro-conservation.”

Table 1, experimental is misspelled.

Reviewer 1 ·

Basic reporting

From round 1 to 2 improvements have been made but significant improvements are needed still, particularly in the introduction.

In my first review the introduction was an area that needed significant refinement. While some changes have been made I still have significant concerns over the organization, flow and content.
- This may be personal preference, so I will let the editor weigh in, but I feel there are several sentences and areas where the tone of the writing is more “preachy” (for lack of a better word) than objective scientific contextualization of the literature or context. For example on line 61 “… great loss to humanity when primates lose their natural habitat.” While I agree whole heartedly with the sentiment, it’s not one I would place, in this context at least, in a scientific paper. If this study evaluated the loss of primate habitat on humanity it would be more objective. Here it seems at worst inappropriate or at best just out of place here.
- Second paragraph beginning line 63: What is the point of this paragraph and how does it tie in/flow your contextualization of paragraph one to paragraph three? Seems unnecessary and out of place. What is your point in this paragraph? A clearly written paragraph should have a specific point to be made, not sure here.
- Third paragraph beginning line 68: This paragraph reads as a list of weakly connected statements. What is the point, how are your bringing in the previous paragraphs to this one and how does this paragraph set up the next one? What is your point in this paragraph? A clearly written paragraph should have a specific point to be made, not sure here.
- Line 68: Jacobson et al 2015 citation is used for justification that “primate sanctuaries are a perfect setting for conservation education programs” but then the second half of the sentences justifies “primates are considered charismatic species” as justification for this using a second citation of Albert et al. 2018. Shouldn’t the Jacobson citation have the justification that you then interpret?
- Line 70: Skibins & Powell, 2013 – not in reference section.
- Paragraph beginning line 74: Similar to points above, this paragraph (which is over a page long) seems to bounce around ideas/points without a clear line of logic. Significant refinement/organization here are needed.
- Paragraph beginning line 74: Time is spent introducing and discussing empathy, but that is not a topic of this paper. Why is it here, what is the point being made and how does that fit into your study questions?
- Paragraph beginning line 74: It strikes me now as I read this, but throughout this paragraph and the previous paragraphs the authors are spending a lot of time emphasizing why primate sanctuaries are a great area for conservation education to occur, however, they also point out not much research on this topic has been done (which is the point of their study). It seems they are using a lot of space in their introduction to tangentially make the point that sanctuaries are perfect despite not having any direct evidence (but using indirect and poorly organized citations to make their point). I do not think the authors need to spend half the introduction arguing primate sanctuaries are a great area for con ed, I think they should spend the introduction discussing what con ed is, why it is valuable, where it has occurred successfully previously, and then in their concluding paragraph set up how sanctuaries fit this model and have their own unique needs for it and thus this study evaluates their effectiveness at it.
- Line 113: all three citations are non-sanctuary studies. Are you citing them as to say here are non-sanctuary studies, thus sanctuaries are under studied? There is likely a better use of citation to make this point.
Discussion is stronger than first round some points that need addressing:
- Line 426: “As an outdoor activity, the visit to a sanctuary increases children’s connection to nature (Kleespies et al. 2022)” This is unclear. Are you saying this citation found that? This citation was not done in a sanctuary so is incorrect. Are you saying this study found that? If so, did you study connection to nature? From methods you have not stated that. And if you are arguing yes more explanation of this is needed and better contextualization for how and why you are using this citation.
- Line 426: “… and can have a major impact on children’s emotions as they have an up-close encounter with the animals (Hacker & Miller 2016)” This was a zoo study, but citing like sanctuary, I can see point attempting to be made here but it is not contextualized/spelled out sufficiently.
- Line 429: “Also, the role of the educator in helping the children better understand why the animals are housed in the sanctuary, the nature of their natural environment, and their life history and behavior (Prokop et al. 2007).” First this sentence is unclear as it is written. Second in terms of your study this is the first and only mention of the role of the educator. What point are you trying to make here about your study and how does it fit into your findings? Also how does this citation support your point? Overall clarification here is needed.
- Beginning Line 440-444: Again, maybe preference, editor please weigh in, but more “preachy” wording that is not directly related to the specifics of this study, not sure if appropriate given it’s a bit of a stretch between study purpose and point of emphasis being made.

Misc. concerns:
- Figure 8 still have graphical output labels.
- Line 174 you say “baccalaureate” and line 195 you say “high school” – are these the same? I would recommend using one or the other label not both.
- Line 369 – redundantly using “%” and word percent. Use %.
- Line 370: uses word percent, use %
- Line 384: You present statistics for grade, but say it is not included in model as a predictor. If you do not include it in model you should not present the results. How did you get results if not in the final model?
- Some additional editing will be needed, recommend another review of wording/punctuation.

Experimental design

No comments, acceptable.

Validity of the findings

Conclusions are acceptable though some points in conclusion could use refinement, see comments in section 1.

·

Basic reporting

The manuscript is much improved, and I thank the authors for their work on the revisions. I have a few suggestions below. The overall writing quality of the manuscript is good, but there are still a few places where the writing is unclear, or some minor errors are listed below. In addition, I would suggest rewording the hypothesis.

ABSTRACT: I might clarify in lines 30- 31 what the experimental control group means. For example, “Questionnaires were either given before (control) or after (experimental) after visits to 3,205 school children…”

INTRO: Line 132-134 “We hypothesized that children of different grades levels attending an educator-led visit at the Mona sanctuary will increase their knowledge about primate species and will show pro-conservation attitudes toward the welfare and conservation of primates.” How is this different from the following line 134- 136 “We also hypothesized that the knowledge of the species and pro-conservation attitudes are influenced by environmental education activities (EEA), gender and grade level” and Line 136-138 “More specifically, we predicted that all participants will demonstrate more pro-conservation responses after participating in the education activities with similar responses among genders.” This prediction seems to contradict the previous hypothesis.

I would suggest you state them based on how you now separated your findings into study 1 and study 2 and the analysis you ran. For example, you could say something like “In the first study, we used a questionnaire to assess children’s knowledge of the species before (control) and after (experimental) EEA. If EEA effectively increases children’s knowledge of the species across grade levels and genders, we predicted that children across grade levels of both genders in the experimental groups will increase accuracy compared to the control group.” This is predicting a main effect (basically your Figure 2). You may also have other a priori predictions regarding grade level and gender since you tested for them. That should also be stated. I would then repeat for Study 2 regarding the attitude survey.

METHODS: Thank you for adding the survey questions to the body of the manuscript; it makes it much easier for the reader to understand. I still maintain the Knowledge questionnaire measure more than knowledge of species but also welfare/ethics and the purpose of MONA sanctuary. Just something to consider when looking at results.

RESULTS: The figures are much improved. In Figure 5, is the check box under Figure 5 incorrect?

DISCUSSION: This still is long and can use some additional editing.

Experimental design

no comment

Validity of the findings

no comment

Additional comments

MINOR ERRORS: I would suggest using a grammar editing program to help catch minor errors and rewording some sentences that are a bit long or awkward.
1. Line 51 insert “of” in front of their
2. Line 94-98 the sentence has several errors and needs to be re-written
3. Line 411 “According to our hypothesis,” should be removed. Your data either supports or doesn’t support your hypothesis which you indicate later in that sentence.
4. Line 413 change effect to affect.
5. Line 430 add “the” in front of conservation
6. Line 441 add a “s” to visit
7. Line 444 use past tense- change to “had”
8. Line 451 change do to does
9. Line 496 change “The way” to How
10. Line 555 add a “s” to survey
11. Please check format for subheadings (inconsistent throughout discussion) and check with Peer J requirements for formatting throughout.

---

## Round 0.3 · accepted · Accept

Thank you for your continued efforts with this manuscript. I think it now reads much more coherently and is ready to be published. However, in proofing, please fix the misspelling of experimental in Tables 1 and 4.

I still think it would be better to use labels for the questions rather than discussing Q7 etc. in the discussion and results as readers would have to keep track of which questions were which.